# Clinical Analysis of Persistent Subretinal Fluid after Pars Plana Vitrectomy in Macula with Diabetic Tractional Retinal Detachment

**DOI:** 10.3390/jcm10245929

**Published:** 2021-12-17

**Authors:** Yong-Koo Kang, Jae-Pil Shin

**Affiliations:** Department of Ophthalmology, School of Medicine, Kyungpook National University, Daegu 41944, Korea; kykyuri@gmail.com

**Keywords:** estimated glomerular filtration rate, persistent subretinal fluid, proliferative diabetic retinopathy, tractional retinal detachment

## Abstract

(1) Background: We analyzed the duration of persistent subretinal fluid (PSF) and the contributing factors of PSF after pars plana vitrectomy in patients who had a macula with diabetic tractional retinal detachment (TRD). (2) Methods: Forty eyes of 40 patients who had pars plana vitrectomy due to a macula with diabetic TRD, between 2014 and 2020, were retrospectively reviewed. The duration of PSF, as well as relevant ocular and systemic factors, was analyzed. (3) Results: The mean duration of PSF was 4.4 ± 4.7 months. The prevalence of PSF was 75.0% at 1 month, 50.0% at 3 months, 30.0% at 6 months and 10.0% at 12 months after surgery. Blood urea nitrogen, creatinine, and estimated glomerular filtration rate (eGFR) were significantly associated with the duration of PSF in the univariate analysis. In the multivariate analysis, only eGFR was significantly associated with the duration of PSF (*β* = −0.089, *p* = 0.030). (4) Conclusion: PSF may persist for more than 12 months in a macula with diabetic TRD after vitrectomy. Moreover, patients with impaired kidney function tended to have a delayed subretinal fluid absorption. Therefore, careful investigation of preoperative systemic conditions, especially kidney function, should be considered before TRD surgery in diabetic patients.

## 1. Introduction

The prognosis of proliferative diabetic retinopathy (PDR) is greatly improved in the era of panretinal photocoagulation and anti-vascular endothelial growth factor (VEGF) intravitreal injection [1,2]. However, tractional retinal detachment (TRD) is one of the most feared complications of PDR and an indication of pars plana vitrectomy in PDR.

Pars plana vitrectomy for TRD may still be one of the most complex vitreoretinal procedures. The goals of this surgery are to remove vitreous hemorrhage, reduce the anterior-to-posterior vitreous traction on the retina, and to relieve epiretinal tangential traction on which proliferative tissue grows. However, iatrogenic retinal breaks are the most major surgical complication, and intraocular tamponade is used to repair retinal breaks.

Persistent subretinal fluid (PSF) was detected via optical coherence tomography (OCT) after the repair of rhegmatogenous retinal detachment [3,4,5], and it was associated with the chronic nature of fluid with a higher viscosity, protein composition and cellularity [6]. Clinical cases of PSF were recently reported after vitrectomy for diabetic TRD [7]. Karimov et al. [8] analyzed the prevalence of PSF in macula with diabetic TRD after vitrectomy. However, all cases received pars plana vitrectomy with intraocular tamponade, and no published report elucidated the effect of intraocular tamponade on PSF in diabetic TRD surgery.

This study reports the prevalence of PSF and analyzes ocular and systemic factors contributing to the duration of PSF after pars plana vitrectomy in patients with PDR accompanying a macula with TRD.

## 2. Materials and Methods

Medical records were reviewed after receiving approval from the Institutional Review Board of Kyungpook National University Hospital (IRB No. 2021-02-009), and the requirement for informed consent was waived because of the retrospective nature of the study. The research was conducted in accordance with the tenets of the Declaration of Helsinki.

### 2.1. Patients

We included PDR patients who had pars plana vitrectomy in macula with diabetic TRD from January 2014 to December 2020. The study included only the patient’s eye that had PDR involving macula with TRD and showed subretinal fluid (SRF) on preoperative OCT. If the patient who had a macula with TRD in both eyes, the eye with poorer visual acuity was used. All patients showed successful anatomical results after vitrectomy and completed at least a 6-month follow-up period with OCT examinations. Patients were excluded according to the following criteria: (1) diabetic TRD without a macula; (2) diabetic TRD without SRF on the preoperative macular OCT scan; (3) the presence of vitreous haze or hemorrhage that is unsuitable for OCT imaging.

Ophthalmic examinations were performed, including the best-corrected visual acuity (BCVA) using a Snellen chart, intraocular pressure (IOP) measurement, slit lamp examination, fundus examination, and OCT examination. All examinations were repeated at baseline and at 1, 2, 4 weeks and every month after vitrectomy. BCVA was converted to the logMAR (logarithm of the minimum angle of resolution) for statistical analyses. The detachment of the macula was confirmed by a preoperative OCT scan. The volume scan of the 6 × 6 mm area was performed using the spectral-domain OCT (Spectralis, Heidelberg Engineering, Heidelberg, Germany). The presence of SRF was defined on the OCT image as a hyporeflective space between the photoreceptor layer and the retinal pigment epithelium (RPE). PSF is defined as SRF present within any area on 6 × 6 mm OCT scans that persists after vitrectomy. Subretinal fluid height (SRFH) was manually measured after vitrectomy by one examiner (YKK) using the built-in ruler in the software of OCT scan. SRFH was measured by determining the vertical distance from the outer interface of the neuroepithelial layer to the inner surface of the RPE in the subfoveal area [9] (Figure 1).

### 2.2. Measurement of Systemic Parameters

Preoperative laboratory testing, including serum glycosylated hemoglobin (HbA1c), blood urea nitrogen (BUN), and creatinine was conducted. The estimated glomerular filtration rate (eGFR) was calculated using the equation recommended by the Chronic Kidney Disease Epidemiology Collaboration [10].

### 2.3. Surgical Technique

A microinvasive pars plana vitrectomy was conducted using 23- or 25-gauge instrument. Phacoemulsification of the cataract with intraocular lens implantation was conducted prior to vitrectomy in phacovitrectomy cases. Vitrectomy was performed using the Constellation surgical system (Alcon, Fort Worth, TX, USA). It included removal of the posterior vitreous and shaving of the peripheral vitreous body; removal of the posterior hyaloid, neovascular, and fibrous epiretinal membranes from the retinal surface with triamcinolone acetonide (TA); and endolaser photocoagulation. If epiretinal membranes with a risk of postoperative re-proliferation were presented, internal limiting membrane (ILM) peeling was conducted after staining with indocyanine green (ICG) dye using membrane forceps. The internal drainage of SRF was conducted if a preexisting or iatrogenic retinal break was found during surgery. For silicone oil (SO) tamponade, if necessary, 5700 centistokes SO (Oxane, Bausch and Lomb, Rochester, NY, USA) was injected into the posterior surface of the pupil after fluid–air exchange. Secondary SO removal surgery was conducted among the SO-filled eyes after confirmation of SRF absorption on OCT scans.

### 2.4. Statistical Analyses

Statistical analyses were performed using the Statistical Package for the Social Science software version 20 (IBM Corp., Armonk, NY, USA). Repeated-measures analysis of variance, corrected by the Bonferroni method, was performed to compare the mean BCVA for the follow-up periods. The Pearson correlation coefficient was used to calculate the mean BCVA at the final visit and the duration of PSF. Ocular and systemic factors associated with PSF duration were analyzed using univariate and multivariate linear regression analysis. Kaplan–Meier analysis was used to show the survival time of PSF; *p*-value < 0.05 was considered significant for all statistical tests.

## 3. Results

### 3.1. Demographics of Patients

One hundred patients were initially included; however, 60 patients were excluded according to the exclusion criteria. Thus, 40 patients (40 eyes) were enrolled in this study. The demographic and clinical characteristics of the enrolled patients are presented in Table 1.

Among the 40 patients, 23 were male and 17 were female. The mean age of all patients was 49.9 ± 12.0 years, the mean interval between diagnosis and surgery was 31.1 ± 32.8 days, and the mean duration of follow-up was 27.3 ± 18.2 months. The duration of diabetes mellitus was 9.2 ± 6.7 years. Fifteen patients had hypertension, and 10 patients had chronic kidney disease following hemodialysis. The preoperative laboratory results revealed that HbA1c was 9.3 ± 2.3, BUN was 24.7 ± 11.2 mg/dL, creatinine was 1.4 ± 0.9 mg/dL, and eGFR was 65.7 ± 29.2 mL/min/1.73m^2^. Thirty-four eyes had received panretinal photocoagulation, whereas 14 eyes received anti-VEGF treatment before surgery. Thirty-three eyes received an intravitreal bevacizumab injection one day before vitrectomy, and thirty-two eyes received phacovitrectomy. During pars plana vitrectomy, ILM peeling was performed in nine eyes. Moreover, thirty-three eyes had a preexisting or iatrogenic retinal break during vitrectomy, and SO tamponade was conducted after fluid–air exchange. Among the 33 eyes, intraoperative SRF drainage was conducted from a retinotomy site in 23 eyes by active aspiration. SO-filled eyes underwent secondary SO removal surgery at 9.6 ± 5.3 months (range, 3.4–22.8 months) after primary surgery.

### 3.2. Duration of PSF and SRFH

The mean duration of PSF was 4.4 ± 4.7 months (range, 0.2–20.5 months). The prevalence of PSF after surgery was 75.0% at 1 month, 50.0% at 3 months, 30.0% at 6 months, and 10.0% at 12 months. The mean SRFH was 106.7 ± 120.7 μm at 1 month, 57.3 ± 91.5 μm at 3 months, 32.3 ± 63.8 μm at 6 months and 5.3 ± 16.4 μm at 12 months after surgery. Figure 2 shows a representative case of PSF on the OCT scans. A Kaplan–Meier plot is used to display the estimated survival probability of PSF (Figure 3).

### 3.3. Best-Corrected Visual Acuity

The changes of mean BCVA were logMAR 1.401 ± 0.653 at baseline, 1.552 ± 0.489 after 1 month (*p* = 1.0), 1.355 ± 0.454 after 3 months (*p* = 1.0), 1.116 ± 0.530 after 6 months (*p* = 0.372), 0.939 ± 0.532 after 12 months (*p* = 0.001), and 0.894 ± 0.695 at the final visit (*p* = 0.003). The mean BCVA significantly improved 12 months after surgery. The mean BCVA at the final visit had no significant correlation with the duration of PSF (r = −0.023, *p* = 0.889).

### 3.4. Clinical Factors Associated with PSF Duration

Univariate and multivariate linear regression analyses of associations between clinical factors and PSF duration are presented in Table 2. BUN (*β* = 0.157, 95% confidence interval (CI) = 0.029 to 0.286, *p* = 0.018), creatinine (*β* = 2.386, 95%; CI = 0.810 to 3.962, *p* = 0.004), and eGFR (*β* = −0.086, 95%; CI = −0.131 to −0.042, *p* < 0.001) were significantly associated with the duration of PSF in the univariate analysis. However, in the multivariate analysis, only eGFR was significantly associated with the duration of PSF (*β* = −0.089, 95% CI = −0.170 to −0.009, *p* = 0.030). There were no significant associations between the duration of PSF and surgical factors, such as internal SRF drainage (*β* = −0.276, 95%; CI = −3.538 to 2.987, *p* = 0.865) or intraocular SO tamponade (*β* = −1.591, 95%; CI = −5.377 to 2.195, *p* = 0.400).

## 4. Discussion

The surgical repair of TRD in patients with PDR is technically challenging. Iatrogenic retinal breaks are most likely to occur because of the friable nature of the ischemic retina and intraocular bleeding commonly occurs during the peeling of fibrovascular membrane [11]. Theoretically, intraocular tamponade is unnecessary in TRD surgery if the traction is relieved surgically without creating retinal breaks [12]. Recently, Tao et al. [13] reported successful clinical results of vitrectomy for TRD in PDR without intraocular tamponade and suggested that intraocular tamponade may be unnecessary for the treatment of TRD in eyes with PDR.

However, there is controversy over intraoperative procedures such as internal drainage or intraocular tamponade during pars plana vitrectomy in TRD. Meredith et al. [12] proposed the technique of membrane segmentation in TRD surgery, stating that there is no need to drain the SRF because the relief of traction is sufficient for allowing the reattachment of the retina, and intraocular tamponade is unnecessary if the traction is surgically released without creating retinal breaks. However, in fact, intraocular SO or gas was frequently used during pars plana vitrectomy for severe TRD cases [12,14,15,16].

Karimov et al. [8] reported that the duration of PSF in diabetic TRD after vitrectomy in eyes with SO tamponade showed significantly faster SRF absorption and intraoperative drainage showed faster SRF absorption, although it is not statistically significant. Thus, they suggested that SO was a favorable option of tamponade in cases with intraoperative breaks and internal drainage. However, intraocular SO tamponade and SRF drainage during vitrectomy are not related to the duration of PSF in the present study (Table 2).

Jung et al. [17] reported the outcomes of ILM peeling on PSF after pars plana vitrectomy in TRD. ILM may serve as a focus of tangential traction caused by migration of fibroblasts and fibrinous materials in diabetic retinopathy. Therefore, they suggested ILM peeling as an effective treatment option for PSF to eliminate the potential traction source and increase the elasticity of the retina, allowing the retina to flatten. However, there was no significant correlation between ILM peeling and the duration of PSF in this study.

Instead, we found that a lower eGFR level was a significant risk factor for long-standing PSF. The absorption of SRF depends on passive diffusion and the active pumping by RPE [18]. However, in diabetic retinopathy, the presence of SRF may be caused by impaired RPE pumping and the disruption of an external limiting membrane, which serves as a barrier to subretinal space and contributes to fluid shifting from the intraretinal space to the outer retina [19]. Moreover, diabetic retinopathy with chronic kidney disease and decreased serum albumin may lower the intravascular osmotic pressure and increase hydrostatic pressure in outer retina or choroid, which could lead to fluid retention and flow into the subretinal space [20,21]. Therefore, we suggested that decreased kidney function could be attributed to delayed SRF absorption.

Given the similar pathophysiological features in diabetic retinopathy and nephropathy, studies found associations between the kidney function with diabetic retinopathy and diabetic macular edema. Man et al. [22] suggested that lower levels of eGFR were associated with the presence and severity of diabetic retinopathy. Tsai et al. [21] also reported the correlation of kidney function and residual SRF after intravitreal, anti-VEGF injection in diabetic macular edema patients. Recently, they also reported that the systemic overhydrating condition was the independent factor for diabetic macular edema [23]. Although there are no reports of SRF after TRD surgery, it is clear that impaired kidney function accompanying diabetic retinopathy inhibits the absorption of SRF. Recently, Takamura et al. [24] reported that dialysis influenced the status of diabetic retinopathy in end-stage renal disease patients, and hemodialysis contributed to the improvement of diabetic retinopathy, including diabetic macular edema. We assumed that the hemodialysis was related to the PSF duration based on their study, but there was no significant correlation (Table 2).

This study has some limitations. It is a retrospective study with a small sample size and lack of a control group. Serum albumin was measured in only small number of patients. Therefore, further studies involving a greater number of cases will be necessary. Nevertheless, this study is significant for analyzing the risk factors of PSF in diabetic TRD patients after surgery.

## 5. Conclusions

In conclusion, we showed that OCT detects PSF after successful diabetic TRD surgery. In addition, patients with impaired kidney function due to diabetic nephropathy tend to have a delayed SRF absorption. Therefore, the investigation of preoperative systemic conditions in PDR patients should be considered before TRD surgery. Additionally, it is important to explain to patients with impaired kidney function that PSF could be long-lasting after having a macula that requires diabetic TRD surgery.

## Figures and Tables

**Figure 1 jcm-10-05929-f001:**
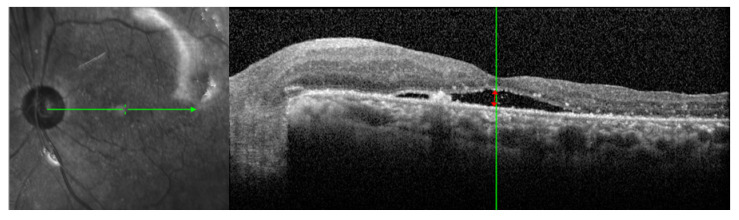
Measurement of subretinal fluid height (SRFH) in OCT scan. SRFH corresponds to the vertical distance from the outer interface of the neuroepithelial layer to the inner surface of the retinal pigment epithelium (red double-headed arrow).

**Figure 2 jcm-10-05929-f002:**
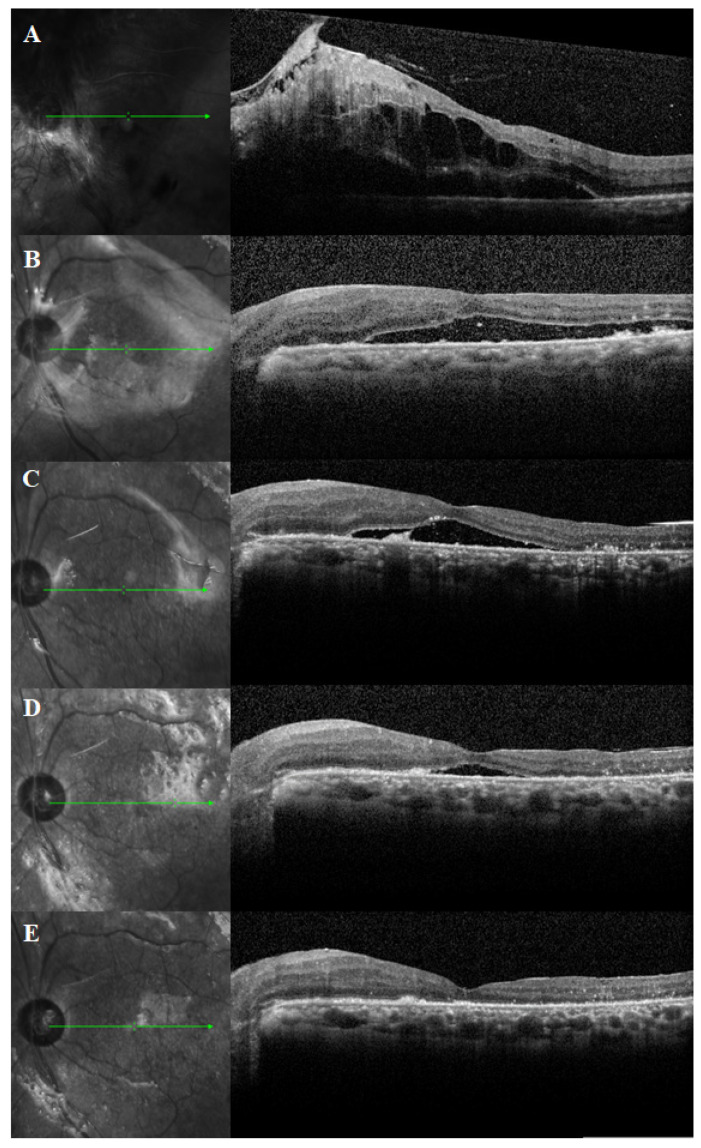
Optical coherence tomography (OCT) scans of diabetic tractional retinal detachment with chronic macular detachment in a 51-year-old male. OCT scans at (**A**) baseline, (**B**) 1 month, (**C**) 3 months, (**D**) 6 months, and (**E**) 12 months after vitrectomy with silicone oil injection.

**Figure 3 jcm-10-05929-f003:**
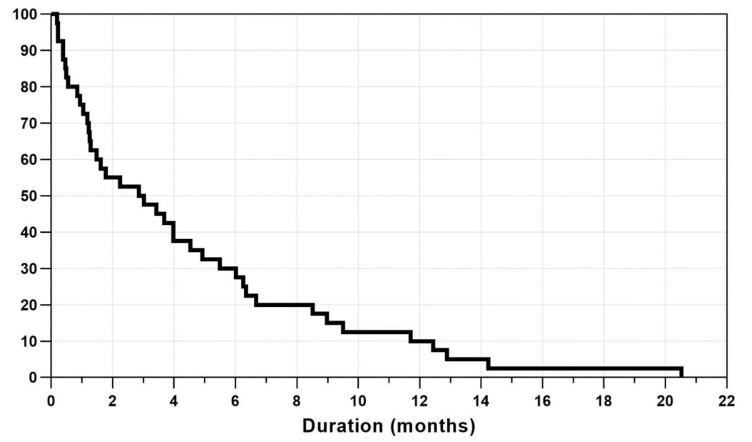
Kaplan–Meier plot illustrating the survival probability of persistent subretinal fluid.

**Table 1 jcm-10-05929-t001:** Demographic and clinical characteristics of enrolled patients.

Characteristics	Value
Number of eyes, *n* (%)	
OD	14 (35.0%)
OS	26 (65.0%)
Sex, *n* (%)	
Male	23 (57.5%)
Female	17 (42.5%)
Mean age, years	49.9 ± 12.0
Mean duration between diagnosis and surgery, days	31.1 ± 32.8
Mean follow up periods, months	27.3 ± 18.2
Duration of PSF, months	4.4 ± 4.7
Systemic characteristics	
Duration of diabetes mellitus, years	9.2 ± 6.7
HbA1c, %	9.3 ± 2.3
Hypertension, *n* (%)	15 (37.5%)
Chronic kidney disease, *n* (%)	10 (25.0%)
BUN, mg/dL	24.7 ± 11.2
Creatinine, mg/dL	1.4 ± 0.9
eGFR, mL/min/1.73 m^2^	65.7 ± 29.2
Ocular characteristics	
Panretinal photocoagulation prior to vitrectomy, *n* (%)	34 (85.0%)
Anti-VEGF therapy prior to vitrectomy, *n* (%)	14 (35.0%)
Preoperative intravitreal bevacizumab injection, *n* (%)	33 (82.5%)
Phacovitrectomy, *n* (%)	32 (80.0%)
Internal SRF drainage during vitrectomy, *n* (%)	23 (57.5%)
ILM peeling during vitrectomy, *n* (%)	9 (22.5%)
Intraocular SO tamponade, *n* (%)	33 (82.5%)

Values are presented as the mean ± standard deviations—BUN: blood urea nitgogen; eGFR: estimated glomerular filtration rate; HbA1c: Hemoglobin A1c; ILM: internal limiting membrane; PSF: persistent subretinal fluid; SO: silicone oil; SRF: subretinal fluid; VEGF: vascular endothelial growth factor.

**Table 2 jcm-10-05929-t002:** Univariate and multivariate analyses of clinical factors associated with the duration of persistent subretinal fluid.

Parameters	Univariate			Multivariate		
	Odds Ratio	CI (95%)	*p*-Value	Odds Ratio	CI (95%)	*p*-Value
Male sex	0.099	−2.993 to 3.192	0.949			
Age at TRD diagnosis	−0.013	−0.142 to 0.115	0.836			
Duration of DM	−0.032	−0.265 to 0.200	0.779			
Hemoglobin A1c	−0.123	−0.884 to 0.638	0.745			
CKD on hemodialysis	−1.456	−4.954 to 2.042	0.405			
BUN	0.157	0.029 to 0.286	0.018	−0.094	−0.343 to 0.156	0.452
Creatinine	2.386	0.810 to 3.962	0.004	1.017	−2.392 to 4.427	0.549
eGFR	−0.086	−0.131 to −0.042	<0.001	−0.089	−0.170 to −0.009	0.030
Axial length	−0.429	−2.361 to 1.504	0.656			
PRP prior to vitrectomy	−0.775	−5.049 to 3.499	0.716			
Anti-VEGF therapy prior to vitrectomy	1.315	−1.861 to 4.491	0.407			
Preoperative intravitreal bevacizumab injection	2.655	−1.273 to 6.583	0.179			
Phacovitrectomy	−0.276	−3.538 to 2.987	0.865			
Internal SRF drainage	−1.445	−5.063 to 2.199	0.430			
ILM peeling during vitrectomy	1.852	−1.758 to 5.462	0.306			
Intraocular SO tamponade	−1.591	−5.377 to 2.195	0.400			

BUN: blood urea nitrogen; CKD: chronic kidney disease; DM: diabetes mellitus; eGFR: estimated glomerular filtration rate; ILM: internal limiting membrane; PRP: panretinal photocoagulation; SO: silicone oil; SRF: subretinal fluid; TRD: tractional retinal detachment; VEGF: vascular endothelial growth factor.

## Data Availability

All the study data are reported in this article.

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
