# Peer review of "Clinical Analysis of Persistent Subretinal Fluid after Pars Plana Vitrectomy in Macula with Diabetic Tractional Retinal Detachment"

_jcm, 2021, doi:10.3390/jcm10245929_

Round 1
Reviewer 1 Report
This is an interesting analysis of factors associated with the persistence of subretinal fluid (SRF) after surgery for macula-off TRD in PDR. The finding and conclusion that this is associated with renal dysfunction, is most interesting, and may be clinically relevant. Was serum albumin measured? Other suggestions for improvement:
- Minor language editing is recommended throughout the manuscript.
- Methods: It seems, according to the Kaplan-Meier graph, that some 20% of the eyes had persistent SRF more than 6 months. Therefore, it is surprising that the minimum required duration of follow-up, for inclusion in the study, was only 3 months (Methods, line 62) . Please explain! Where there any drop-outs, in whom the SRF persisted until last follow-up? How was their data managed?
- It is not clear exactly where in the macula that the height measurement of the SRF was obtained. Was only the maximum height considered, anywhere along the 6x6 mm scan? Or was only the "subfoveal" height considered? Related: Please include the "en face" image of the fundus in Figure 1, along with the cross-sectional OCT image, so that it will be obvious for the reader, where exactly the OCT line was projected on the fundus, ie where the measurement was obtained.
- Persistent SRF may be due to persistent traction on the retina. Triamcinolone Acetonide (TA) may be useful to visualize persistent vitreous after having completed the membrane dissection. Persistent vitreous may lead to membrane reproliferation, recurrence and traction, and therefore may affect the duration of the persistent SRF. It was not stated whether intraoperative TA was used in this regard. Was the membrane dissection carried out bi-manually, with the assistance of chandelier light, or was the dissection performed uni-manually?
- 1st sentence of Discussion is inserted by error, please delete it.
- There is no control group. Mention this among the limitations of the study.
Author Response
We appreciate your comments. According to your review, we revised the sentence.
This study is a retrospective chart review and preoperative laboratory testing including renal parameter, such as BUN and creatinine, was performed for general anesthesia before surgery. However, we couldn’t confirm serum albumin which was not included the laboratory testing.
1. We know to need language correction of the sentence and revise the sentence. However, it’s difficult to make language editing because of short term of period manuscript revision. Therefore, we will conduct additional language editing after revision of manuscript.
2. There was a mistake in the process of writing the sentence. All included patients were followed up at least 6 month, and patients under 6 month of follow up were excluded in this study. Therefore, the sentence in the method section was revised to 6 month.
3. We measured the maximal height of SRF on the 6 x 6mm OCT scan. According to your proposal, we added cross sectional OCT and en-face image with OCT scan line as Figure 1 that explains the measurement of SRF height and added en-face image with OCT scan line as Figure 2.
4. In vitrectomy, it was confirmed in all cases that vitreous membrane was removed after membrane dissection using TA. Membrane dissection procedure was performed uni-manually, but if needed, surgery was performed bi-manually using chandelier illuminations. In addition, ICG dye staining was performed in the case of ILM peeling procedure.
5. There was a mistake in the process of writing the sentence. We delete 1st sentence of discussion section.
6. As your comment, this study has a limitation that control group could not be determined because of the nature of this study. We added the context in the discussion section.
Reviewer 2 Report
Authors summarized patients with impaired kidney function tended to have delayed subretinal fluid absorption.
The paper is well written and the study has a well-structured design. There are some minor concerns to be addressed.
In the discussion section, authors should delete first paragraph.
This part is not appropriate for this manuscript.
"Authors should discuss the results and how they can be interpreted from the perspective of previous studies and of the working hypotheses. The findings and their implications should be discussed in the broadest context possible. Future research directions may also be highlighted."
Author Response
We appreciate your comments. According to your review, we revised the sentence.
First of all, we delete first paragraph of discussion section.
Moreover, we added discuss this result and further research directions in discussion section of paragraph.
Reviewer 3 Report
Dear authors,
thank you for your effort in conducting this study on retinal detachments in diabetes patients. I think, you address a clinically relevant question.
However, I have some inquiries and suggestions.
First of all, english language should be improved. Some sentences are even hard to understand at all (e.g. ll. 59, 103, 128, 193, 195, 204).
Please check again, whether it should be macula instead of macular in the title and the rest of the manuscript.
The introduction is fine and summarizes all relevant information.
In the methods, please clarify whether the follow-up mode was used. The choice of methods and their weaknesses should be discussed in the end. Especially, since manual measurements were made (one or two examiners?) it is prone to errors.
Please also discuss, how OCT scans can be conducted when the macula is off and how SRF measurements are made in these cases.
Also for the discussion: you have a very heterogeneous group of patients (with or without cataract surgery, with or wothout oil ....). This makes it hard to conclude anything from your data.
How did you ckeck for normality? Did you correct for multiple testing?
How was the postoperative management, how were follow-ups planned?
All in all, the discussion is very generalized. You should focus much more on your data and discuss them more in detail.
Thank you for your work and kind regards.
Author Response
We appreciate your comments. First of all, we revised the sentence of lines according to your comment. In addition, we will conduct additional language editing after revision of manuscript.
We checked the word “macular” was corrected. Therefore, we revised the sentence with the word “macular”.
SRF height was manually measured by one examiner using the built-in ruler in software program. According to your comment, it could be prone to error. However, the measurement was made as consistently as possible based on the method of reference 9. Moreover, all enrolled patients were macular off status in preoperative OCT scans, thus, manual measurement was conducted using the built-in ruler in software program. We added this content in method section.
The data were analyzed after the normality test and made corrections for multiple analyses if not satisfied normal distribution.
The subject patients are receiving treatment appropriately for complications other than TRD, especially diabetic macular edema.
Discussion may be limited because there is not yet research on the effect of PSF in diabetic retinopathy. However, based on this study, we suggested the effect of renal dysfunction on PSF and we added the sentence of further investigation in diabetic retinopathy in the discussion section.
Round 2
Reviewer 3 Report
Dear authors,
thank you for your reply.
I do not think you addressed my comments to an adequate extent and there are more grammatical errors than before.
Kind regards.
Author Response
Point 1: (Round 2)
Dear authors,
Thank you for your reply.
I do not think you addressed my comments to an adequate extent and there are more grammatical errors than before.
Kind regards.
Response:
We appreciate your comments. Sorry for the grammatical error and we tried to correct it as much as possible.
We revised again the whole manuscript according to your recommendations.
We have requested professional manuscript editing, and we will make language correction again after revisions.
Thank you for your reply again.
Point 2: (Round 1)
Please check again, whether it should be macula instead of macular in the title and the rest of the manuscript.
Response:
We requested the correct notation of the term "macular" and "macula" and got the answer the term "macula involving TRD" or "macular OCT" were correct from language editing service.
So, we revised the words again.
